# Managing an Invasive Weed Species, *Parthenium hysterophorus*, with Suppressive Plant Species in Australian Grasslands

**DOI:** 10.3390/plants9111587

**Published:** 2020-11-16

**Authors:** Amalia Belgeri, Ali Ahsan Bajwa, Asad Shabbir, Sheldon Navie, Gabrielle Vivian-Smith, Steve Adkins

**Affiliations:** 1School of Agriculture and Food Sciences, The University of Queensland, Gatton, QLD 4343, Australia; abelgeri84@gmail.com (A.B.); ali.bajwa@dpi.nsw.gov.au (A.A.B.); s.adkins@uq.edu.au (S.A.); 2INIA La Estanzuela, Ruta 50, Colonia 70000, Uruguay; 3Weed Research Unit, New South Wales Department of Primary Industries, Wagga Wagga, NSW 2650, Australia; 4School of Life and Environmental Sciences, The University of Sydney, Camden, NSW 2570, Australia; 5IVM Group Pty. Ltd., Varsity Lakes, QLD 4227, Australia; snavie@ivmgroup.com.au; 6Biosecurity Queensland, Department of Agriculture and Fisheries, Ecosciences Precinct, Brisbane, QLD 4102, Australia; Gabrielle.Viviansmith@deedi.qld.gov.au

**Keywords:** invasive alien species, parthenium weed, species diversity, pastures, weed management, suppressive plants

## Abstract

Parthenium weed has been invading native and managed Australian grasslands for almost 40 years. This study quantified the potential of selected plant mixtures to suppress the growth of parthenium weed and followed their response to grazing and their impact upon plant community diversity. The first mixture consisted of predominantly introduced species including Rhodes grass, Bisset bluegrass, butterfly pea and green panic. This mixture produced biomass rapidly and showed tolerance to weed species other than parthenium weed. However, the mixture was unable to suppress the growth of parthenium weed. The second mixture of predominantly native pasture species (including forest bluegrass, Queensland bluegrass, Buffel grass and siratro) produced biomass relatively slowly, but eventually reached the same biomass production as the first mixture 12 weeks after planting. This mixture suppressed parthenium weed re-establishment by 78% compared to the control treatment. Its tolerance to the invasion of other weed species and the maintenance of forage species evenness was also superior. The total diversity was five times higher for the mixture communities as compared to the plant community in the control treatment. Therefore, using the suppressive pasture mixtures may provide an improved sustainable management approach for parthenium weed in grasslands.

## 1. Introduction

Invasive alien plant species negatively affect ecosystem structure and functioning at multiple tiers [1,2]. These species often disturb the native species composition and reduce the vegetative biodiversity by dominating the landscape of the invaded area. Parthenium weed (*Parthenium hysterophorus* L.) is such an invasive weed species, having immense ecological and agricultural impacts [3]. Originating from the tropical and subtropical Americas, this noxious invader has now infested more than 40 countries around the world, including in Africa, Asia, the Pacific and Australia [4]. It is a major weed in crops and pastures, acts as an alternative host to several major crop pests, negatively affects native plant species diversity within a wide range of environments, causes toxicity problems to domesticated livestock and significant health problems to people [5,6,7,8,9,10,11,12]. Parthenium weed is an aggressive invader due to its unique biological, physiological and ecological adaptive features [5,13,14].

Parthenium weed has infested an area of more than 60 million ha in Queensland, Australia [3]. Adamson [15] reported losses worth $69 million (AUD) per annum in the beef industry due to parthenium weed. Different management approaches have been used to control parthenium weed with varying degrees of success. Several chemical herbicides provide effective control of this species. However, the large scale weed infestations and the environmental concerns rule out the continuous and frequent use of chemicals. Ecologically based approaches have been proposed as more sustainable to manage this species [16,17,18]. In Queensland, a biological control program for parthenium weed was started in the late 1970s and since then, nine insect and two fungi biocontrol agents have been released into that state [19,20]. One of the most successful biological control agents is the leaf-feeding beetle (*Zygogramma bicolorata* Pallister; Coleoptera: Chrysomelidae) [21]. Although this agent has widely spread within the parthenium weed infestation range, it does not provide a remarkable control alone [18]. Therefore, the search for alternative non-chemical methods to control parthenium weed has remained a key theme for weed scientists in this part of the world.

Another important management approach is the use of competitive/suppressive pasture plant species to control parthenium weed. The use of competitive crop cultivars or improving the crop competition to manage weeds have proved successful in Australian crop production systems [22]. However, relatively little research has been carried out to explore the potential of suppressive plant species with a secondary fodder value to control environmental and/or grassland weed species such as parthenium weed [23]. Previous studies in Australia have shown that several valuable pasture grass and legume species such as Bisset bluegrass (*Bothriochloa insculpta* Hochst. ex A. Rich A. Camus), Floren bluegrass (*Dichanthium aristatum* Poir. C.E. Hubb), Buffel grass (*Cenchrus ciliaris* L.) and butterfly pea (*Clitoria ternatea* L.), when sown alone, can successfully suppress the growth of parthenium weed under glasshouse conditions [23,24]. The Bisset bluegrass, Floren bluegrass, Buffel grass, Rhodes grass (*Chloris gayana* Kunth) and Queensland bluegrass (*Dichanthium sericeum* (R. Br.) A. Camus) were able to suppress the growth of parthenium weed under field conditions [23,25]. However, for a practical application of an approach using suppressive plants, the use of only one species will be unable to provide the best quality fodder as compared to using a mixture of species and will not contribute to the diversification of the grassland agroecosystem [26,27]. Additionally, the sowing of a single species might pose a long-term threat as the repeated sowing of the same species might lead to that plant itself becoming invasive [28].

The benefits of increasing species diversity within a grassland-based agroecosystem will include not only the creation of a higher and more stable fodder biomass production but also improved forage quality and palatability [26]. Other benefits might also include better recycling and retention of soil nutrients, improved soil physical properties and increased community resilience and resistance to further weed invasion [26,29,30]. Thus, increasing a grassland’s diversity, especially if achieved with desirable native plants, should be one of the main objectives and ultimately influence the pasture management recommendations. It is also important to note that suppressive plant species may act synergistically with the biological control agents of parthenium weed [17,18,31,32]. Therefore, the potential of suppressive plants must be explored in a broader integrated management context.

This study evaluated the potential suppressive effect of two selected plant mixtures against parthenium weed growth in grazed and non-grazed situations. The impact of these suppressive plants on plant community diversity was also assessed. The specific objectives were to a) compare the suppressive ability of two selected plant mixtures, one predominantly composed of native species and the other predominantly composed of introduced species, against parthenium weed under field conditions, b) determine the fodder biomass production of these two suppressive plant mixtures, and c) assess the impact that these two suppressive plant mixtures had upon the plant community diversity.

## 2. Results

### 2.1. Suppression of Growth and Abundance of Parthenium Weed

The parthenium weed populations forming within the two sown plant mixtures were of equal biomass by 30 days after emergence (DAE) as compared to that seen in the control plots, with no obvious suppression of biomass detected (Figure 1a). By 60 DAE, however, the weed biomass had been significantly reduced (*p* < 0.05) in both plant mixtures and the control in comparison to 30 DAE and remained low as the season progressed (Figure 1a). Unlike biomass production, the abundance of parthenium weed was not identical within the two plant mixtures at 30 DAE (9.8, 8.8 and 4.8 plants m^−2^ in the control, mixture one and mixture two plots, respectively; Figure 1b). By 60 DAE, the abundance of parthenium weed had been reduced in both grassland plant mixtures and the control as the season progressed (2.6, 5.5 and 1.7 plants m^−2^; Figure 1b) and by 90 DAE the abundance had become significantly reduced (*p* < 0.05) in both plant mixtures, but more in mixture two (78% reduction with respect to the control, 0.44 plants m^−2^; Figure 1b).

### 2.2. Biomass Production of the Plant Mixtures

At 30 DAE, the unsown control treatment produced a significantly greater forage species dry biomass (35.2 g m^−2^, *p* < 0.05; Figure 2) than either of the two sown plant mixtures (20.5 and 17.5 g m^−2^) as the grassland plants were still developing. However, by 60 DAE, all treatments were growing actively and mixture one produced the greatest dry biomass recorded (189.3 g m^−2^; *p* < 0.05). The number of sown plant individuals contributing to this biomass was also significantly higher for mixture one than for mixture two when measured at 60 DAE (14.8 and 11.1 plants m^−2^, respectively; Table 1). Not all species within the plant mixtures established well. From mixture one, the major contribution to biomass production was made by Rhodes grass, butterfly pea and green panic, all giving good establishment rates of 5.9, 4.8 and 3.1 plants m^−2^, respectively (Table 1), with Bisset bluegrass and bull Mitchell grass showing poor establishment (Table 1). From mixture two the major contribution to biomass production was made by siratro (Macroptilum artropurpureum (DC.) Urb.) and Buffel grass with plant establishment rates of 4.9 and 4.1 plants m^−2^, respectively, with red grass and Hoop Mitchell grass not establishing well (Table 1). Both plant mixtures produced significantly more biomass than the control at 90 DAE (Figure 2).

### 2.3. Below- and Above-Ground Species Composition and Diversity

At 30 DAE, prior to the sowing of the plant mixtures, the seed bank analysis showed that the below-ground community was dominated by sedges (e.g., slender flat-sedge, *Cyperus gracilis* R.Br; 6172 viable seeds m^−2^; 41.2% of the total), parthenium weed (1969 viable seeds m^−2^; 13.1% of the total) and other broadleaf species (shady wood-sorrel, *Oxalis exilis* A. Cunn; 1031 seeds m^−2^; 6.8% of the total; Table 2). From the grass species, the most dominant, according to abundance, was blue couch (*Digitaria didactyla* Wild.; 1000 seeds m^−2^; 6.6% of the total).

The multivariate analysis showed no similarity (i.e., no overlap) between the species composition of the above-ground community created by plant mixture one and the control (Figure 3) whereas similarity existed between plant mixture two and the control. When pooling the data from all three vegetation surveys, plant mixture one showed the greatest species composition stability (i.e., similar species composition over time; R = 0.68, *p* < 0.001). However, the unsown species were those that contributed the most to this similarity score (Table 3). The sown species green panic (*Panicum maximum* Jacq.) and unsown species such as green crumbweed (*Dysphania carinata* (R.Br.) Mosyakin & Clemants) and goosegrass (*Eleusine indica* (L.) Gaertn.) contributed the most to the dissimilarities between the control and plant mixture one (dissimilarity 55.4%). The sown species siratro and Buffel grass and the unsown species parthenium weed, spiny amaranth and goosegrass contributed the most to the dissimilarities between the control and mixture two (similarity percentage (SIMPER) dissimilarity 52.4%).

The community diversity including the sown species was highest at 30 DAE for plant mixture one, then declined as the season progressed, possibly as short-lived annuals completed their life cycle (H’ 2.3; *p* < 0.05; Figure 4a). The forage species diversity, which overall was lower than the total community diversity, showed the same trend (Figure 4c). On the other hand, when the sown species were excluded there was a significantly lower diversity indicating that the sown species enhanced total and forage species diversity (*p* < 0.05 and *p* < 0.01, respectively). There were no significant differences between treatments at any of the sampling periods in the forage species diversity (Figure 4b,d). Maximum evenness (J’) was recorded at 60 DAE for both sown mixtures (0.9; *p* < 0.01; Figure 5). Forage species evenness was not different for both sown mixtures at 30 and 60 DAE but both were greater than the control (*p* < 0.01; Figure 5). Mixture one recorded a significant reduction in evenness at 90 DAE (0.311; *p* < 0.01), and at this point in time the control treatment achieved a similar forage species evenness J’ to mixture two (0.7 and 0.8, respectively; Figure 5).

## 3. Discussion

The selected pasture species had a significant suppressive effect on parthenium weed growth and biomass production. The variable performance of the two mixtures could be attributed to the growth and suppressive effects of individual species present in each mixture. For instance, the presence of siratro and Buffel grass in the second plant mixture contributed towards the effectiveness of this mixture in reducing the abundance of parthenium weed as both species have been reported to grow vigorously and suppress parthenium weed individually [35]. Moreover, these species were found to be tolerant to the allelopathic effects of parthenium weed [36,37]. Siratro could play an important role in the suppression of parthenium weed growth through its twining and prostrate growth habit, creating an enveloping canopy that would shade the early growth of the weed. The formation of an early canopy has been indicative of species that could successfully suppress the growth of parthenium weed [25]. Rapid growth and attainment of height and branching or tillering, are some of the morphological and physiological characters of grassland plants that have also been positively correlated with the ability to suppress parthenium weed growth [35] and would thus explain the successful suppression of mixture two. In addition, Queensland bluegrass, the only native species that established well in this plant mixture (Table 2), could also play a decisive role in reducing the abundance of parthenium weed as it has been shown to suppress parthenium weed by as much as 50% when in monoculture in the field [35]. Therefore, this species may have played a role in this trial as well.

The effective suppression of parthenium weed abundance in the field was not only due to the suppressive nature of plant mixture two but also its interaction with a biological control agent, the leaf-feeding beetle [18]. Occasionally, the biological control agents are quite damaging; however, not all years are climatically advantageous for this insect to act as effectively as it did in 2011. Thus, having competitive perennial species to control parthenium weed over the longer term becomes an important part of the overall strategy. At 30 DAE, the biomass from newly established pasture from both seed mixtures was relatively low and was not yet able to sustain grazing (Figure 2), indicating that, in this environment, an establishment period (with the exclusion of grazing) is necessary. The well below average rainfall during November 2011 also contributed to this slow establishment which therefore may have been more rapid in a typical rainfall year.

Perennial grass-based pastures such as the two used here are known to establish more slowly than annual pastures, however after establishment they are usually very resilient to grazing [14]. Indeed, both plant mixtures (native and introduced) achieved effective ground cover by 60 DAE, even after having initial low stand establishment rates, possibly because of the advantageous larger seed size than their counterpart grass species. Larger seeds can produce stronger seedlings that survive and thus establish better [38]. Most native species in the plant mixtures (i.e., bull Mitchell grass, Hoop Mitchell grass and red grass) established poorly, which reduced the richness of native species expected in plant mixture two. This result suggests a poor adaptation of the *Astrebla* genus to the microclimate of this region. Improving the germination and establishment rates of these native species should become a research priority in order to gain benefit from their inclusion in plant mixtures.

At 60 DAE, the greater production of biomass by pasture mixture one (Figure 2) suggested that this mixture consists of species that are all capable of rapid growth and probably better suited to the present study environment. For instance, Rhodes grass is an abundant but introduced species in the region (D. Youles personal communication, 2010). The number of established plants from the plant mixtures that contributed to their biomass was also significantly higher in plant mixture two than one. Additionally, mixture one showed less variability on plant heights throughout the period (Table 1). The peak in biomass production achieved by mixture one also indicates that this mixture responded better to the above average rainfall during January (236.6 mm). At 90 DAE, the below average rainfall during February (87 mm) resulted in the control treatments yielding significantly less biomass than was seen at 60 DAE. However, both pasture mixtures were able to maintain a stable biomass production over time and even after simulated grazing had been applied (Figure 2). The high abundance of species with good drought tolerance in both pasture mixtures probably contributed to this response [16].

Across vegetation surveys and the susceptibility of the mixtures to weed invasion, the lack of overlap detected for mixture one by the non-metric multidimensional scaling (NMDS) analysis was, to an extent, expected to occur for mixture two. Buffel grass, present in mixture two, can act as an environmental weed [28,39], and has been proven to displace native species in central Queensland, Western Australia and the Northern Territory [39]. It has, therefore, been one of the most controversial species to be considered for the suppressive management of parthenium weed. However, the value of Buffel grass as a pasture species is well recognized and, as yet, it is not a declared weed in Queensland. From plant mixture one, only green panic became a dominant sown species. The remaining species responsible for the lack of overlap were the unsown broadleaves with no grazing value such green crumbweed, which became more abundant in this mixture than in the control. Mixture one also recorded the lowest forage species evenness at 90 DAE, indicating the domination of just a few, or even a single, species at this time (Figure 5). At the third simulated grazing time green panic was the only sown species present. This evolution towards a monoculture is certainly not desirable. The dominance of Buffel grass in pasture mixture two was anticipated and therefore a decrease in the evenness was expected. However, this mixture maintained both a steady total and forage species evenness throughout the study period.

Certain weed species add to the vegetation diversity and therefore they are considered a positive component in ecological terms, but they are frequently associated with negative effects upon domestic livestock production [26]. The similarity in species composition between mixture two and the control was certainly due to the unsown ”weedy” liverseed grass (*Urochloa panicoides* Beauv.) and other unsown broadleaved weed species, such as Paddy’s Lucerne, purslane small-flowered mallow (*Malva parviflora* L.) and thorn apple (*Datura ferox* L.), which were equally abundant in both mixture two and the control.

The calculation of forage species diversity separately from the total community and also excluding the sown species allowed an assessment of the sown species upon the diversity of unsown species with forage significance. The total diversity was ca. five times higher for the mixture communities than for the forage control community, indicating that the contribution of unsown species with grazing value in this trial was not substantial. This observation was also supported by the pre-trial seed bank analysis, which showed a very high proportion of broadleaf and sedge species to be present at the site (Table 2). The better plant establishment achieved by mixture one at 60 DAE may have produced a greater total community diversity. However, when sown species were excluded from the analysis no differences were observed between this and the other treatments (Figure 4), indicating that the plausible benefits on increasing diversity of species with a grazing value from the sown mixtures has not yet occurred. Positive species interactions, such as an improvement in the abundance of native species with grazing value, were expected to occur. However, three months of field evaluation might have limited these kinds of observations as in sub-tropical Queensland, changes in community composition due to a sown pasture may take 10 or more years [6,16]. In the longer term, nitrogen fixation by the legume component, better nutrient recycling and a more balanced usage of the available resources might result in obtaining positive responses in yield, forage quality and community species diversity.

The significant decline in diversity noted at 90 DAE for pasture mixture one was probably due to the increased dominance of only a few species in that mixture (i.e., green panic and the unsown weed species green crumbweed and bull thistle (*Cirsium vulgaris* (Savi) Ten)). The suppressive ability of one species over another may vary significantly across environments [40]. Other studies, testing these same species in a monoculture, have shown higher biomass production can be achieved and therefore they exhibit better suppressive ability against parthenium weed [23]; however, the species were evaluated under higher fertile soils and over longer periods of time. Total yield is not the sole criterion for evaluating the benefits of increased forage species diversity on grazing lands. As demonstrated in this investigation, the ability to suppress the reestablishment of an invasive species and the contribution to plant diversity are valuable characters of sown plant mixtures. This study also demonstrated that managing pastures to increase forage species diversity is a useful component to introduce into an integrated weed management scheme for parthenium weed.

## 4. Materials and Methods

### 4.1. Study Location

This field study was conducted in the Kilcoy district (26.57° S, 152.30° E, 189 m above sea level) of south-east Queensland, Australia, from September 2011 to March 2012. The topography of the site was gently sloping with good drainage and consisted of a typical soil for the region (i.e., a brown-grey dermosol, pH 6). The characteristic vegetation of the region was originally a native grassland that has been under continuous grazing for at least 100 years and infested with parthenium weed for at least 25 years (D. Youles, personal communication 2011). This grazing practice has led to the replacement of certain desirable native species such as black speargrass (*Heteropogon contortus* L.) with other less desirable species such as green couch (*Cynodon dactylon* (L.) Pers.). The majority of the species present within these grasslands are perennials with a high presence of forbs and graminoids [8,14]. A two-strata vegetation system has developed, as is commonly found in long-term grazed, sub-humid grasslands, and consists of a low dense stratum, no more than 5 cm tall, and a taller stratum of bunched grasses and small woody plants [41]. The leaf-feeding beetle (*Z. bicolorata*) has been present in south-east Queensland for at least 25 years and present at the Kilcoy site for at least 20 years.

The climate at the site is sub-tropical, with a long-term average annual precipitation of 959 mm, occurring mainly in the summer months of January and February. Before and during this study, the annual rainfall was well above average in 2010 (1619 mm), above average in 2011 (1168 mm) and close to average in 2012 (980 mm) [42]. The mean temperatures for the region ranged from 7 ℃ for night-time lows in winter to 29 ℃ for day-time highs during summer [42]. Before the study was undertaken, the site was grazed by cattle at a typical stocking rate for the Kilcoy district of ca. 0.5 cows ha^−1^ during the drier winter months (June to August) and 0.8 cows ha^−1^ during the wetter summer months (December to February). Just prior to use, the site was cleared of vegetation using a bulldozer fixed with a front blade, then cultivated three times to a depth of ca. 20 cm using a disc cultivator (Connor Shea 18 Disc Series, Connor Shea Napier Pty Ltd., Melbourne, Victoria). The site was then protected by a wire fence that prevented unwanted entry of cattle and wildlife for the duration of the study.

### 4.2. Experimental Design and Treatments

The experiment was performed using a Latin square fully randomized design within a total study site area of ca. 225 m^2^. Each of the two sown plant mixtures was replicated three times with a net plot size of 3 m × 3 m. Moreover, three unsown plots were maintained as control plots. Within each 3 × 3 m treatment plot, a permanent quadrat (1 m × 1 m) was marked out in the middle, from which plant heights could be calculated at each of the simulated grazing times. A 1 m wide path was maintained around each treatment plot, and a 2 m wide path was maintained around the study site and inside the fence line to avoid any “edge effects” within three randomly placed quadrats (each 1 m^2^).

The plant mixtures were created by considering the following criteria: the species were previously shown to be suppressive of the growth of parthenium weed (either ranked as strongly or moderately suppressive) [23], species providing functional group complementation (e.g., providing a stabilizing effect upon forage production and quality [43], species known to have good germination, seedling establishment and growth in the Kilcoy district and species recommended by the local experts. Ultimately, seed lot availability was the defining factor determining the final mixture composition (Table 4).

All plant seed lots used in the mixtures were obtained from seed companies, then cleaned, air dried and stored at 15 ± 1 ℃ and 15 ± 2% relative humidity (RH) in a seed store before being used. In order to use an appropriate sowing rate for each species in the field, a laboratory germination test was undertaken on a sub-sample of the seed lot in a germination incubator (day/night temperature regime 25/20 ± 2 ℃ with a 12 h day/12 h night photoperiod and light intensity of 100 µmol m^−2^ s^−1^), mimicking the environmental conditions in the field at the peak time for germination of parthenium weed [7]. The germination value was then used to modify the recommended sowing rates (Table 4).

Sowing of pre-weighed plant mixtures was undertaken in late November 2011 immediately after the last disc ploughing occasion. To achieve an even sowing rate in each treatment plot the seed mixes were added to dry sawdust (5 g of sawdust to each 1 g of seed mix), mixed well, then evenly spread by hand over the appropriate treatment plot. Next, the seed mix was covered by hand-raking of the soft soil surface, and was then compacted under foot to ensure good seed-to-soil contact. Several unsown control plots were marked, consisting of the vegetation established after the last disc ploughing.

### 4.3. Weed Population

To estimate the size of the parthenium weed population across the site and prior to the sowing of plant mixtures, the soil seed bank was determined from five soil cores (each of ca. 7.2 cm in diameter and 10.0 cm deep) collected from the four corners and the centre of three randomly placed quadrats (1 m^2^ each). The five samples from each quadrat were then mixed, providing a total of three random samples taken from over the whole study site. A sub-sample from each soil seed bank sample of ca. 600 g was were then spread thinly (*ca*. 5 mm) over a sterilized soil mixture (University of California mixture; forming a 3 cm thick basal layer) that was contained within shallow plastic germination trays (each 20 × 25 × 6 cm, w/L/h; one soil sample per tray). These trays were then placed at random onto a bench in a shade house, wetted to field capacity and then analyzed for emerging seedlings over a 6 month period to allow for all of the species in the seed bank to be identified, including those with long-term seed dormancy. Three control trays of the sterilized soil mixture were placed among the experimental trays to monitor for any seedlings that may have arisen from this mixture or from the glasshouse environment. Once they emerged, seedlings were identified and counted, removed and discarded. In the case where immediate identification was not possible, representative individuals were planted into small pots of compost and grown to maturity, to allow for later taxonomic identification. When no further emergence was recorded in the seedling trays for a period of 2 weeks (ca. 26 weeks after the start of the study), the soil was allowed to dry for 1 week, stirred, then rewetted to trigger further germination.

### 4.4. Data Collection

Once the plants had established and were growing rapidly, three simulated grazing times were made at 30, 60 and 90 days after emergence (DAE). The simulated grazing times consisted of cutting (with scissors) both the sown and unsown grass species at a point that was equal to 50% of their average height. The average heights were determined from a permanent quadrat that had been placed in the middle of each the various treatment plots. Those quadrats that had been cut on a previous simulated grazing time were not re-cut at 60 and 90 DAE. Upon cutting, all plant material was divided into either parthenium weed, sown species or all other unsown grass species. These samples were put into separate paper bags and dried in an oven at 70°C for 3 days. Species considered to be “weeds” (for instance, all unsown broadleaved species, such as Paddy’s Lucerne (*Sida rhombofolia* L.), purslane (*Portulaca oleracea* L.) and spiny amaranth (*Amaranthus spinosus* L.) were not included in the biomass samples taken, as it was assumed these species would not be grazed. Just prior to each of these simulated grazing interventions, species identification and abundance were also determined.

### 4.5. Statistical Analysis

On each occasion prior to the simulated grazing, species diversity within the above-ground communities was assessed using the Shannon–Weiner index (1), both for the whole plant community (*H*′ total) and for the species that were considered to be forage species consumed by domestic livestock (*H*′ forage species):*H*′ = −∑*^S^_i_*_= 1_*p_i_*log*_e_p_i_*(1)
where *S* is the number of species or species richness and *p_i_* is the relative abundance of each species, calculated as the proportion of individuals of a given species to the total number of individuals in the community
*P_i_* = *n_i_*/*N*(2)
where *n_i_* is the number of individuals in species *i* (i.e., the abundance of species *i*) and *N* is the total number of all individuals [44]. The evenness within the above-ground vegetation was assessed using the Pielou’s evenness index (2), both for the whole community (*J*′ *total*) and for the forage species (*J*′ *forage species*):*E* = *H*′/*H*_max_(3)
where *E* is the equitability of species whose value ranges from 0 to 1, *H*′ is the observed species diversity and *H*_max_ is the maximum species diversity, *H*_max_ = log*_e_S* [44,45].

Patterns of plant species composition and temporal changes in response to the treatments were determined using non-metric multidimensional scaling (NMDS) in two dimensions on both species abundance and presence/absence data, based on a Bray–Curtis dissimilarity matrix. The extent of clustering according to the plant mixtures was assessed using a maximum of 5000 permutations in an analysis of similarity (ANOSIM), an analogue of the univariate analysis of variance (ANOVA) specifically developed for ecological data, which tests for differences between and within groups of samples (multivariate) from different times and treatments, and does not require normality and homoscedasticity assumptions, generating the statistic R. Values of R range from −1 to +1, with values approaching R = +1 indicating a strong dissimilarity among samples. When differences between treatment plots at α < 0.05 or lower were detected, a ”similarity percentage” routine (SIMPER) was used in order to identify which species were primarily contributing to the similarities/dissimilarities among treatments. Species abundance was subjected to square root transformation to enhance their fit to the models. Both abundance and presence or absence data sets gave similar results and so only those from square-root-transformed abundance data are presented. All multivariate analyses were performed using the software Primer (Version 6.0; Primer Technologies, Inc., San Francisco, CA, USA).

Data sets of species diversity (*H*′) and evenness (*J*′) and biomass production were analyzed by running an ANOVA using a general linear model procedure in Minitab, version 16 (Minitab Inc., PA, USA). The general linear model was set up with the three simulated grazing times (30, 60 and 90 DAE) and three grassland plant treatments (control, seed mixture 1 and seed mixture 2) as the main factors for the analysis of the species diversity and evenness indexes and biomass variables. Analysis was undertaken using an adjusted sum of squares approach using 95.0% confidence intervals. A transformation of the data set was required when the assumptions of the ANOVA were not met (e.g., species abundance and parthenium weed biomass). However, for all cases the significance of the test did not change, therefore for practical purposes the results are presented as the original values.

## 5. Conclusions

Plant mixture one could be characterized as an early biomass producer as it is able to rapidly respond to changes in environmental conditions and is more tolerant to invasion by species considered ”weeds” other than parthenium weed. On the other hand, plant mixture two can be characterized as a later biomass producer, but achieving the same biomass levels as mixture one, and able to suppress parthenium weed re-establishment. We have demonstrated that managing pastures to increase forage species diversity is a useful component to introduce into an integrated weed management scheme for parthenium weed. The mixture to be selected depends on the main objectives of the production system.

## Figures and Tables

**Figure 1 plants-09-01587-f001:**
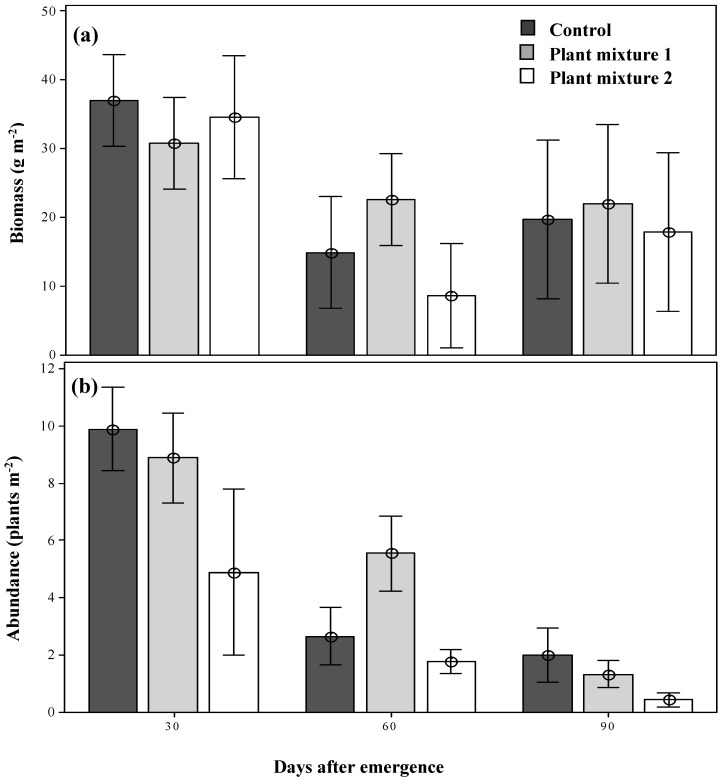
The dry biomass (**a**) and the abundance (**b**) of parthenium weed under the two plant mixture control treatments at three simulated grazing times. Error bars represent ± standard error of means.

**Figure 2 plants-09-01587-f002:**
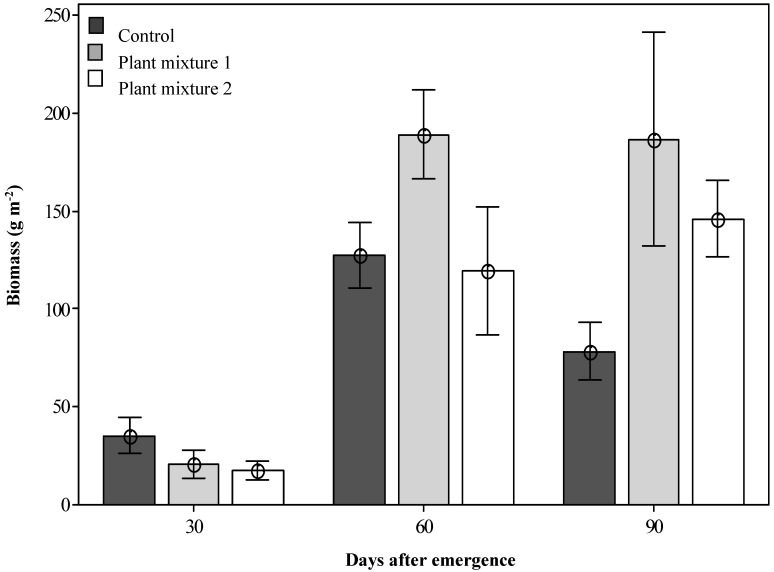
The dry biomass produced by the two sown plant mixtures (forage species only) at three simulated grazing times after emergence. Error bars represent ± standard error of means.

**Figure 3 plants-09-01587-f003:**
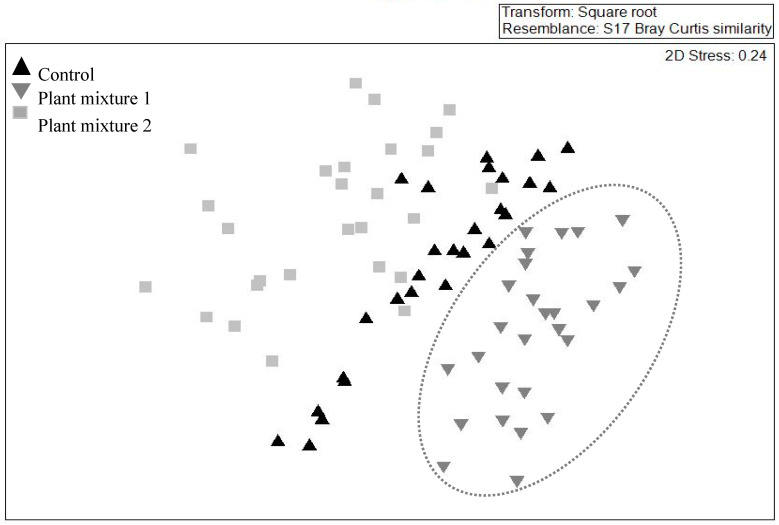
Effect of two plant mixtures (treatments) and the unsown control on the above-ground plant community. The illustration was drawn by non-metric multidimensional scaling in two dimensions on abundance data, based on a Bray–Curtis dissimilarity matrix and across three survey occasions. Note mixture one data (
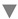
) are enclosed as only this mixture did not overlap with the control.

**Figure 4 plants-09-01587-f004:**
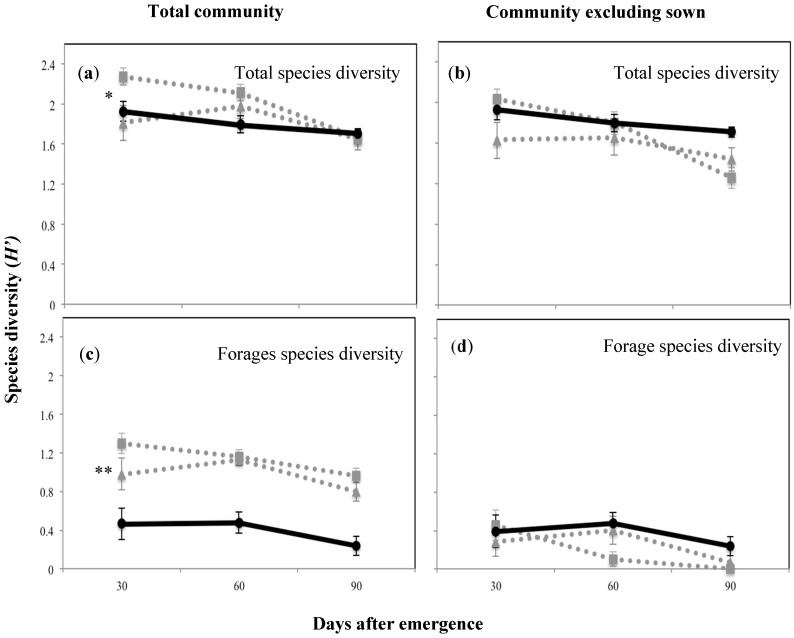
Total and forage species diversity expressed as the Shannon–Weiner index (H’) for the above-ground plant community and at the three survey occasions of the plant mixtures and the control treatment, where (**a**) shows the total species diversity including sowed species; (**b**) shows the species diversity excluding sowed species; (**c**) shows the forage species diversity including sowed species; (**d**) shows the forage species diversity excluding sowed species. Solid black line with circles shows control treatment, while dotted grey lines with squares and triangles show mixture one (Rhodes grass + creeping bluegrass + butterfly pea + green panic) and mixture two (forest bluegrass + Queensland bluegrass + siratro + Buffel grass), respectively.* *p* < 0.05; ** *p* < 0.01.

**Figure 5 plants-09-01587-f005:**
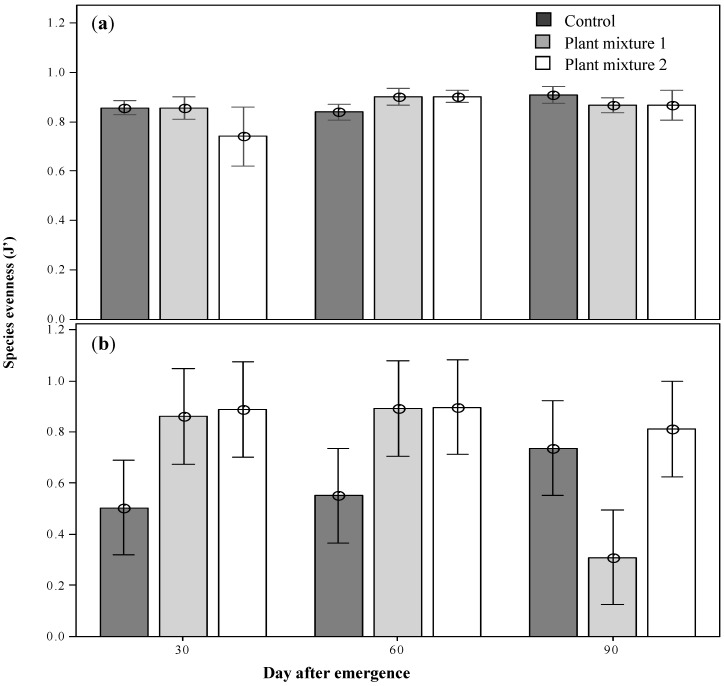
The above-ground total community evenness (**a**) and the above-ground forages evenness (**b**) the plant mixtures at the three survey occasions after emergence. Means ± 2SE.

**Table 1 plants-09-01587-t001:** The establishment and heights of the different sown plant mixtures at the three survey occasions (means ± SEM).

Pasture Mixture	Plant Species	Plant Establishment at the Three Survey Occasions (DAE; Plants m^−2^)			Height at the Simulated Grazing Times (cm)		
		30	60	90	30	60	90
One	Rhodes grass	5.6 ± 0.8	5.9 ± 0.8	6.6 ± 0.3	50.6 ± 6.8	98.7 ± 4.5	131.7 ± 8.1
	Bisset bluegrass	4.0 ± 1.2	1.0 ± 0.0	1.0 ± 0.0	27.6 ± 8.6	63.0 ± 1.0	88.5 ± 3.3
	Butterfly pea	2.8 ± 0.4	4.8 ± 0.6	1.8 ± 0.3	15.2 ± 1.3	45.3 ± 4.8	67.8 ± 4.9
	Green panic	3.3 ± 0.9	3.1± 0.5	4.9 ± 1.1	43.5 ± 9.2	114.9 ± 3.4	143.7 ± 8.0
	Total	15.7 ± 0.9	14.8 ± 1.1	14.3 ± 1.2	-	-	-
Two	Forest bluegrass	4.4 ± 1.3	1.0 ± 0.0	2.0 ± 0.0	18.7 ± 1.7	53.5 ± 0.5	112.5 ± 3.5
	Queensland bluegrass	5.8 ± 0.9	1.3 ± 0.1	1.0 ± 0.0	28.3 ± 1.9	60.2 ± 4.0	102.3 ± 23.0
	Siratro	2.1 ± 0.4	4.1 ± 0.4	3.4 ± 0.6	14.5 ± 0.3	54.8 ± 2.4	65.6 ± 7.9
	Buffel grass	4.3 ± 0.7	4.9 ± 0.7	6.3 ± 0.8	25.18 ± 1.9	103.0 ± 3.0	147.9 ± 67.3
	Total	16.6 ± 1.0	11.1 ± 0.9	12.7 ± 1.0	-	-	-

Bull Mitchell grass and red grass did not establish properly out of mixtures one and two, respectively.

**Table 2 plants-09-01587-t002:** The composition of the soil seedbank of the study site infested with parthenium weed prior to sowing of the plant mixtures in November 2011.

Family	Species	Status ^1^	Germinable Seeds (m^−2^)
Amaranthaceae	*Alternanthera nana* R.Br	W	141
	* *Amaranthus spinosus* L.	W	234
Apiaceae	* *Cyclospermum leptophyllum* (Pers.) *Sprague*	W	63
	* *Soliva* sp.	W	16
Asteraceae	* *Gamochaeta pensylvanica* (Willd.) *Cabrera*	W	63
	* *Cirsium vulgare* (Savi) Ten.	W	31
	* *Conyza bonariensis* (L.) Cronq.	W	63
	* *Conyza sumatrensis* (Retz.) E.H. Walker	W	16
	* *Crassocephalum crepidioides* (Benth.) S Moore	W	16
	* *Gamochaeta americana* (Mill.) Wedd	W	16
	*** *Parthenium hysterophorus* L.**	**W**	**1969**
Brassicaceae	* *Lepidium africanum* (Burm.f.) DC.	W	1000
	* *Lepidium bonariense* L.	W	63
	* *Lepidium didymum* L.	W	625
Campanulaceae	*Wahlenbergia stricta* (R. Br.) Sweet	W	63
Chennopodiaceae	*Dysphania carinata* (R.Br.) Mosyakin & Clemants	W	78
	*Dysphania pumilio* (R.Br.) Mosyakin & Clemants	W	31
	*Einadia trigonos* (Schult.) Paul G. Wilson	W	47
Crassulaceae	*Crassula sieberiana* (Schult. & Schult.f.) Druce	W	47
Cyperaceae	* *Cyperus brevifolius* (Rottb.) Hassk.	W	531
	*Cyperus gracilis* R.Br.	W	6172
	*Cyperus iria* L.	W	16
Fabaceae	* *Medicago polymorpha* L.	D	23
Gentianaceae	* *Schenkia spicata* (L.) Mansion	W	141
Iridaceae	* *Sisyrinchium* sp. Peregian	W	47
Malvaceae	* *Sida cordifolia* L.	W	31
	* *Sida rhombifolia* L.	W	31
Oxalidaceae	*Oxalis exilis* A. Cunn.	W	1031
	*Oxalis purpurea* L.	W	0
Plantaginaceae	*Plantago debilis* R.Br.	W	47
Poaceae	*Chloris divaricata* R.Br.	D	94
	* *Chloris gayana* Kunth	D	94
	* *Cynodon dactylon* (L.) Pers	D	78
	*Digitaria didactyla* Willd.	D	1000
	* *Eleusine indica* (L.) Gaertn.	W	141
	*Paspalidium distans* (Trin.) Hughes	D	234
	*Sporobolus creber* De Nardi	D	47
	*Sporobolus elongatus* R.Br.	D	375
	* *Urochloa panicoides* Beauv.	W	16
Polygonaceae	*Rumex brownii* Campd.	W	16
Portulacaceae	* *Portulaca oleracea* L.	W	250

* Introduced species in Australia. ^1^ Status: W = weed (undesirable) and D = desirable.

**Table 3 plants-09-01587-t003:** The contribution of each species to the community similarities across three survey occasions for each plant mixture, their life form and status in Australia ^1^.

Pasture Mixture	Plant Species		Life Form ^2^	Status ^3^	Contribution (%)
	Common Name	Scientific Name			
**Control (similarity 59.5%)**	**Liverseed grass**	***Urochloa panicoides* Beauv.**	G	W	16.9
	Paddy’s lucerne	*Sida rhombifolia* L.	Sb	W	15.0
	Parthenium weed	*Parthenium hysterophorus* L.	H	W	12.7
	Bala	*Sida cordifolia* L.	Sb	W	12.3
	Goose grass	*Eleusine indica* (L.) Gaertn.	G	W	11.4
	Spiny amaranthus	*Amaranthus spinosus* L.	H	W	11.3
	Green crumbweed	*Dysphania carinata* (R.Br.) Mosyakin & Clemants *	H	W	6.3
	Blue couch	*Digitaria didactyla* Willd. *	G	D	3.5
	Spiked malvastrum	*Malvastrum coromandelianum* (L.) Garcke	Sb	W	2.7
One (similarity 68.2%)	Spiny amaranthus	*Amaranthus spinosus* L.	H	W	12.8
	Green crumbweed	*Dysphania carinata* (R.Br.) Mosyakin & Clemants *	H	W	11.9
	Paddy’s lucerne	*Sida rhombifolia* L.	Sb	W	11.5
	Parthenium weed	*Parthenium hysterophorus* L.	H	W	11.0
	Green panic	*Panicum maximum* Jacq.	G	D	10.1
	Butterfly pea	*Clitoria ternatea* L.	H	D	8.3
	Rhodes grass	*Chloris gayana* Kunth	G	D	5.8
	Spear thistle	*Cirsium vulgare* (Savi) Ten.	H	W	5.3
	Bala	*Sida cordifolia* L.	Sb	W	4.4
	Bisset bluegrass	*Bothriochloa insculpta* Hochst. Ex A.Rich A.Camus	G	D	3.6
	Yellow vine	*Tribulus micrococcus* Domin *	H	W	2.8
	Purslane	*Portulaca oleracea* L.	H	W	2.7
Two (similarity 55.4%)	Siratro	*Macropitilum artropurpureum* (DC.) Urb.	H	D	16.2
	Buffel grass	*Cenchrus ciliaris* L.	G	D	13.5
	Liverseed grass	*Urochloa panicoides* Beauv.	G	W	11.3
	Green crumbweed	*Dysphania carinata* (R.Br.) Mosyakin & Clemants *	H	W	11.1
	Paddy’s lucerne	*Sida rhombifolia* L.	Sb	W	10.6
	Spiny amaranthus	*Amaranthus spinosus* L.	H	W	8.6
	Bala	*Sida cordifolia* L.	Sb	W	7.3
	Parthenium weed	*Parthenium hysterophorus* L.	H	W	5.8
	Yellow vine	*Tribulus micrococcus* Domin *	H	W	3.7
	Goosegrass	*Eleusine indica* (L.) Gaertn.	G	W	3.0

^1^ Higher similarity values indicate consistency in species composition over time. ^2^ Life form (G = grass, H = herb, Sb = subshrub) and ^3^ weed status (W = weed, D = desired) were determined by the characteristics described in the literature [33,34]. * Native species.

**Table 4 plants-09-01587-t004:** The plant species present in the treatment mixtures, their suppressive indices, and field suppression percentages and seed rates *.

Plant Mixture	Common Name	Scientific Name	Family	Origin	SI *	FS (%) *	Seed Rate (seed m^−2^)
One	Rhodes grass	*Chloris gayana* Kunth *cv.* Callide	Poaceae	Introduced	-	79	245
	Bisset bluegrass	*Bothriochloa insculpta* Hochst. ex A. Rich A. Camus *cv.* Bisset	Poaceae	Introduced	3.2	62	33
	Butterfly pea	*Clitoria ternatea* L. *cv*. Milgarra	Fabaceae	Introduced	2.9	70	10
	Green panic	*Panicum maximum* Jacq.	Poaceae	Introduced	-	-	118
	Bull Mitchell grass	*Astrebla squarrosa* C.E. Hubb.	Poaceae	Native	1.4	62	9
Two	Forest bluegrass	*Bothriochloa bladhii* (Retz.)	Poaceae	Introduced	2.0	-	38
	Red grass	*Bothriochloa macra* (Steud.) S. T. Blake	Poaceae	Native	0.9	-	10
	Queensland bluegrass	*Dichanthium sericeum* (R. Br.) A. Camus	Poaceae	Native	1.7	70	41
	Siratro	*Macroptilum artropurpureum* (DC.) Urb.	Fabaceae	Introduced	-	-	65
	Buffel grass	*Cenchrus ciliaris* L.	Poaceae	Introduced	1.8	76	100 **
	Hoop Mitchell grass	*Astrebla elymoides* F. Muell.	Poaceae	Native	1.1	-	38

* The suppressive index (SI) and field suppression (FS) ability over parthenium weed (%) were identified in previous field studies [24,25], and the adjusted seeding rates were calculated using the germination percentage obtained from each seed lot and the recommended seeding rates. ** Scarified seeds.

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
