# Peer review of "Managing an Invasive Weed Species, Parthenium hysterophorus, with Suppressive Plant Species in Australian Grasslands"

_plants, 2020, doi:10.3390/plants9111587_

Round 1

Reviewer 1 Report

You have provided a good summary for a potential method of reducing an invasive weed of your region. Developing non-chemical practices for controlling weeds is a very important area of research and will benefit numerous animal and crop producers. The study provides extensive details of plant species in the research plots and different methods for analyzing diversity. Although plant mix 2 may provide some weed suppression, it is difficult to make a definitive statement based on 1 field study from a single location and over only 1 growing season. I suggest you expand the study for future evaluations in order to make a more impactful statement to whether a specific mix is effective. Specific comments are included in the attached file.

Reviewer 2 Report

Dear authors,

My recommendations to minor corrections are as follows:
1) Line 474, 478, 483, 489, 492, 494, 499, 506, 522 - Please, add comma after the year of publication
2) Latin names of plant species should be in italic - Please, check and correct the names in the Reference section
3) The author of the latin name of plant species should not be in italic - Please, check and correct this in Table 3

I have just one note to your work  - the experiment has no repetition in time, i.e. 2 or 3 consequent years. 

Best regards
